# Bibliometric Analysis of the Knowledge Base and Future Trends on Sarcopenia from 1999–2021

**DOI:** 10.3390/ijerph19148866

**Published:** 2022-07-21

**Authors:** Yao Xiao, Ziheng Deng, Hangjing Tan, Tiejian Jiang, Zhiheng Chen

**Affiliations:** 1Department of Endocrinology, Endocrinology Research Center, Xiangya Hospital of Central South University, Changsha 410008, China; yaoxiao0433@csu.edu.cn (Y.X.); jiangtj@tom.com (T.J.); 2Institute of Reproduction and Stem Cell Engineering, School of Basic Medical Science, Central South University, Changsha 410008, China; d.ziheng@csu.edu.cn (Z.D.); hjtan2017@csu.edu.cn (H.T.); 3Centers of System Biology, Data Information and Reproductive Health, School of Basic Medical Science, Central South University, Changsha 410008, China; 4National Clinical Research Center for Geriatric Disorders, Xiangya Hospital, Changsha 410008, China; 5Department of Pediatrics, The Third Xiangya Hospital, Central South University, No. 138 Tongzipo Road, Yuelu District, Changsha 410013, China

**Keywords:** sarcopenia, bibliometric analysis, CiteSpace, VOSviewer

## Abstract

Sarcopenia is characterized by progressive loss of muscle mass and function, and it is becoming a serious public health problem with the aging population. However, a comprehensive overview of the knowledge base and future trends is still lacking. The articles and reviews with “sarcopenia” in their title published from 1999 to 2021 in the SCIE database were retrieved. We used Microsoft Excel, VOSviewer, and CiteSpace to conduct a descriptive and bibliometric analysis. A total of 3582 publications were collected, from 4 published in 2000 increasing dramatically to 850 documents in 2021. The USA was the most productive country, with the most citations. The Catholic University of the Sacred Heart and Landi F were the most influential organization and author in this field, respectively. The core journal in this field was the *Journal of Cachexia Sarcopenia and Muscle*. According to the analysis of keywords and references, we roughly categorized the main research areas into four domains as follows: 1. Definition and diagnosis; 2. Epidemiology; 3. Etiology and pathogenesis; 4. Treatments. Comparing different diagnostic tools and the epidemiology of sarcopenia in different populations are recent hotspots, while more efforts are needed in the underlying mechanism and developing safe and effective treatments. In conclusion, this study provides comprehensive insights into developments and trends in sarcopenia research that can help researchers and clinicians better manage and implement their work.

## 1. Introduction

Sarcopenia is a common geriatric syndrome characterized by progressive loss of skeletal muscle mass and function [1]. According to a recent meta-analysis, the global prevalence of sarcopenia varies between 10% to 27% depending on different diagnostic criteria. Meanwhile, the prevalence of severe sarcopenia ranges from 2% to 9% [2]. Sarcopenia is also associated with an increasing risk of falls, fractures, disability, and even death [3,4]. More healthcare expenditure is related to sarcopenia, no matter the community, perioperative, or general hospital settings [5]. For patients combined with chronic diseases, such as liver cirrhosis [6], dialysis [7], and chronic obstructive pulmonary disease (COPD) [8], sarcopenia is associated with a higher prevalence and poorer prognosis. As the global population aging trend intensifies, sarcopenia can seriously affect the health of older adults and cause a huge personal, social, and economic burden [1].

Given the magnitude of the problem, the field of sarcopenia research is gaining increasing attention worldwide. In the past few years, corresponding diagnostic criteria and consensus have been developed for different ethnic groups [9,10,11,12,13]. In 2016, sarcopenia was assigned a separate ICD-10-CM code [14]. Currently, a large number of original research articles and reviews have focused on the etiology, diagnosis, prevention, and treatment of sarcopenia. However, traditional systematic reviews have limitations in generalizing an overall view of a specific field over a large time frame [15]. Bibliometric is a kind of systematic, objective, and repeatable analysis of the literature by using mathematical and statistical methods [16,17], presenting a landscape of knowledge bases, hotspots, and emerging trends [18,19]. A recent study used a topic search strategy and retrieved more than 13,000 publications related to sarcopenia between 2001 and 2020, suggesting that the research trends will continue to focus on aging, nutrients, and molecular mechanisms [20]. Susan et al. [21] analyzed the top 100 cited articles published before 2019, and recognized the most significant contributions and scientific breakthroughs in sarcopenia research. Tan et al. [22] discussed the relationship between muscular atrophy/sarcopenia and cardiovascular disease in the elderly using bibliometric ways. However, the search strategy used for bibliometric analysis can greatly affect the results, and the articles and reviews with “sarcopenia” in the title, which are believed to be more representative of the core content in this field [23], are often overlooked. Moreover, how the entire field of sarcopenia has evolved and the current status of sarcopenia research is not fully addressed in previous studies [20,21]. In this study, we searched for articles and reviews with “sarcopenia” in the title. Descriptive and bibliometric analyses were performed to reveal the knowledge base, hotspots, and mainstream domains in the field of sarcopenia. We also attempted to reveal the evolution and future trends of the entire field using timeline analysis and burst detection of co-cited documents. Our study aims to provide an overview of sarcopenia research and new clues to help researchers and clinicians plan and manage their future work.

## 2. Materials and Methods

### 2.1. Database Selection 

Web of Science Core Collection (WOSCC, Clarivate Analytics, Philadelphia, PA, USA) is considered one of the most frequently used databases for bibliometric studies. We conducted our bibliometric study relying on the Science Citation Index Expanded (SCIE, 1999-), a subdatabase of WOSCC. SCIE database provides the bibliometric data including titles, authors, countries, institutions, abstracts, keywords, journals, and especially reference information, for download and further analysis.

The 2020 impact factor and quartile in the category of journals were acquired from Journal Citation Reports (http://clarivate.com/products/web-of-science, accessed on 20 February 2022). Data on gross domestic product (GDP) in 2021 and the percentage of the population aged 65 or above were obtained from the World Bank (http://data.worldbank.org/indicator, accessed on 9 April 2022). Ethical approval was not needed for the bibliometric study.

### 2.2. Search Strategy

The search strategy we used was that TI = sarcopenia, document types were restricted within articles and review articles, and the publication year was between 1999 to 2021. It is worth mentioning that we did not restrict language in our search. Figure 1 illustrates the literature search, selection, and analysis process. To avoid bias, we conducted this search on 16 February 2022. A total of 3582 documents were retrieved for data analysis.

### 2.3. Data Standardization

All bibliometric records of the 3582 retrieved literature were downloaded as “full record and cited reference” from SCIE. Microsoft Excel 2019 (Microsoft Corporation, Redmond, WA, USA), CiteSpace version 5.8.R3 (Drexel University, Philadelphia, PA, USA) [24], and VOSviewer version 1.6.18 (Leiden University, Leiden, The Netherlands) [25] were used to standardize, analyze, and visualize the data.

Due to the inherent flaws of WOSCC, bibliometric records need to be standardized before use. CiteSpace was used to remove the potential replicates records first. Before formal analysis, we checked and combined the information from each region into their affiliated country, for example, assigning the publications from England, Scotland, Wales, and Northern Ireland to the UK. Since a particular author or institution can be presented with different names and lead to miscalculation, we manually screened the names of identified authors and institutions with high publication volume and merged the information after confirming from the same author or institution. Frequent appearing keywords were assessed and synonyms, such as “DXA” and “dual-energy X-ray absorptiometry”, were incorporated. Finally, a thesaurus_terms.txt file was prepared to reduce redundancies.

### 2.4. Bibliometric Analysis 

Descriptive analyses of bibliometric indicators, including the annual number of publications, countries, authors, institutions, journals, keywords, and citations, were analyzed in Microsoft Excel 2019.

VOSviewer was used to conduct the coauthorship analysis and keywords co-occurrence analysis, and the normalization method was LinLog/Modularity. Coauthorship analysis of countries, organizations, and authors was used to help uncover the core contributors to the sarcopenia field. Since authors will carefully select the keywords of their papers, and Keywords Plus were generated from the titles of an article’s references [26], we conducted a co-occurrence analysis of both keywords to reveal the hot spots in sarcopenia research. In networks, nodes represent different elements including countries, organizations, authors, and keywords. The total link strength (TLS) is a good indicator of the cooperative or co-occurrence strength. The color indicates the average appearing year (AAY). 

CiteSpace was utilized to perform: (1) Dual-map overlay of journals of the citing and cited references. The links represent the citing pathways, and the thickness reflects the citation strength; (2) Document co-citation analysis, in which cited references were represented by nodes, tree rings were used to represent the citation history of references in each year, and the purple ring represented high betweenness centrality, indicating that the node was more connected by other nodes or was centrally located between different groups of nodes [18]. Cited references were grouped into different clusters when some references were co-cited more frequently than others, and the references in clusters were considered the knowledge base for research in the corresponding subfield. Citing papers were considered as the research fronts, and the titles, keywords, and abstract were extracted to generate the cluster labels using the log-likelihood ratio algorithm. Meanwhile, the network was transformed into a timeline view to better understand the evolution process of each cluster; (3) Burst-detection analysis, and references with strong citation bursts were marked to analyze the hot spots and research trends in different periods. 

### 2.5. Statistical Analysis 

We conducted our statistical analysis with SPSS-26 (IBM Statistical Package for Social Sciences, Chicago, IL, USA). The relationship between the two variables was assessed using the Spearman correlation coefficient and *p* values < 0.05 were considered significant.

## 3. Results

### 3.1. Descriptive and Co-Occurrence Analysis

In total, 2855 (79.70%) articles and 727 (20.30%) reviews among 3582 documents matching the criteria were retrieved. The publications increased dramatically from a total of 4 (0.11%) documents in 2000 to 850 (23.73%) documents in 2021.

A total of 79 countries contributed to sarcopenia publications. Due to some publications being cooperated by different countries, we counted them in each participating country. The top 10 most productive countries are displayed in Appendix A. The country with the highest number of publications is USA (690, 19.26%), followed by China (534, 14.91%) and Japan (524, 14.63%). In terms of total citations of publications, USA is also in a leading position with more than 40,000 times of citations. When adjusted with the GDP, South Korea ranks first with 194.15 papers per trillion GDP. As Appendix A shows, the number of publications on sarcopenia is positively correlated with GDP (*r* = 0.851, *p* < 0.001). Meanwhile, the correlation between the number of publications and the percentage of the old population is low (*r* = 0.338, *p* = 0.003). There are 34 countries that published equal to or more than 20 documents, and these countries were selected to conduct coauthorship analysis (Figure 2). Accordingly, the USA has a high TLS with close collaborations with other countries such as Italy, China, and the UK. The AAY of China is 2019, suggesting sarcopenia research in China grew rapidly recently. 

Appendix A presents the top 10 most prolific organizations, and the Catholic University of the Sacred Heart (92, 2.57%), University of Melbourne (73, 2.04%), and Sichuan University (65, 1.81%) are the top three among 3685 organizations. After excluding two nodes without connection to others, the coauthorship analysis was conducted with 58 institutions with a minimum number of 20 publications (Figure 3). Sichuan University, Wenzhou Medical University, and National Center for Geriatrics and Gerontology are shown as big yellow nodes, indicating that they are emerging research institutions. Noteworthy, the Catholic University of the Sacred Heart, University of Erlangen Nuremberg, and University of Verona are the top three with the most citations and the only three with more than 10,000 citations; this may be partly due to their involvement in the preparations for European Consensus [1,13].

A total of 17,587 authors were identified. Appendix A lists the most influential authors with high publications and citations. Productive authors are from organizations with high publications, 4 of the top 10 most published authors are from the Catholic University of the Sacred Heart. We identified 98 authors who published at least 10 papers. After removing 13 nodes that were not linked to others, the coauthorship analysis is shown in Figure 4. Landi F had the most documents, citations, and the highest TLS. Authors with earlier AAY started earlier in sarcopenia research and are labeled with purple colors, and authors labeled by yellow color may be the emerging influential investigators. 

The documents in the field were published in 851 journals, and the top 10 of the most active are listed in Table 1. The *Journal of Cachexia Sarcopenia and Muscle* (*n* = 117), *Journal of Nutrition Health and Aging* (*n* = 108), and *Nutrients* (*n* = 100) published the most papers concerning sarcopenia research. The *Journal of Cachexia Sarcopenia and Muscle*, with an impact factor (IF) of 12.91 in 2020, is the only one with an IF of more than 10 among the top 10 journals and may be the most influential journal in this field. Dual-map overlay of all literature in the sarcopenia field shown in Figure 5 was constructed by CiteSpace to present the citing trajectory in the entire datasets [27]. The pathway represents the citation relationship between the citing journals on the left and the cited journals on the right. As we can see, sarcopenia research was mainly located in the categories of Molecular, Biology, Immunology, as well as Medicine, Medical, Clinical. The cited journals were chiefly from the categories of Molecular, Biology, Genetics, as well as Health, Nursing, Medicine, indicating they were the intellectual base of sarcopenia research.

Keywords were carefully selected to reflect the core content and theme of the papers. A total of 7367 keywords were retrieved from 3582 documents. After excluding the meaningless keywords and merging words with the same meaning, 116 keywords with a minimum occurrence of 50 times were used to perform co-occurrence analysis (Figure 6). Terms with relatively earlier AAY were colored in purple, such as “age”, “postmenopausal women”, and “resistance exercise”. In the last 3 years, a lot of investigators have paid attention to “consensus”, “sarc-f”, “validation”, “survival”, and “postoperative complication”, indicating they were the research hot spots in recent years and may still flourishing soon. Appendix A showed the distribution of the top 25 keywords with the highest occurrences. The most common keywords, such as “muscle mass”, “obesity”, “body-composition”, etc., are closely related to the clinical characteristic of sarcopenia patients or the epidemiology of sarcopenia. 

### 3.2. Documents Co-Citation Analysis

We list the top 10 most cited papers in Appendix A; the most cited one is the European consensus on sarcopenia published in 2010 [13], with more than 6200 citations. The revised European consensus in 2019 [1] ranked second and was cited more than 2600 times. It is worth noting that “Low relative skeletal muscle mass (sarcopenia) in older persons is associated with functional impairment and physical disability” published in 2002 had citations more than 1800 times, probably because it estimated the prevalence of sarcopenia in American old adults and confirmed the association between reduced muscle mass and functional impairment and disability [28].

To reveal the development and mainstream domains of sarcopenia in recent years, the landscape view generated by document co-citation analysis is based on publications between 2010 and 2021 (Figure 7A). The top 50 most cited publications in each year were selected to build a network of co-cited references that year, and then we merged all individual networks. The final network contains 405 nodes and 491 links. “Sarcopenia: revised European consensus on definition and diagnosis” has received considerable citations since its publication in 2019, as indicated by the largest yellow circle in Figure 7A. The co-citation network was separated into 20 clusters consisting of different groups of cited references; the modularity and weighted mean silhouette of the network are 0.8787 and 0.9651, respectively. The individual silhouette of each cluster is greater than 0.88, which is considered high and indicates the credibility of the network and homogeneity within the clusters. Cluster #0 physical disability is the largest cluster, followed by #1 liver cirrhosis and #2 slow walking. As shown in the timeline view in Figure 7B, the research trend and hot spots shifted from “physical disability”, “myocyte apoptosis”, “low mean mass”, and “relative muscle mass” to “liver cirrhosis”, “slow walking” and “preoperative sarcopenia”.

Burst-detection analysis captures papers with a sharp increase in citations over a specific period, and it helps to find important milestone papers in the development of a research field. References with the strongest citation bursts in the sarcopenia field between 2010 to 2021 were identified (Figure 8). The first of these references was written by Rolland Y et al. [29], who in 2008 reviewed the assessment, etiology, pathogenesis, and consequences of sarcopenia, which provided the basis for subsequent consensus. In addition to the studies we mentioned earlier, Martin L et al. [30] analyzed muscle loss in patients with solid tumors using CT images and found that regardless of whether patients presented obese or not, low muscle index and low muscle attenuation were independently a poor prognostic of survival. While the citation burst for most references has ended, some articles continue to be cited frequently, suggesting that they are still considered reliable sources in recent times. Of these, Shafiee G et al. [31] conducted a meta-analysis and estimated the global prevalence of sarcopenia in healthy adults aged over 60 years of age. The studies they analyzed were conducted between 2009 to 2016 and showed that 10% of men and 10% of women, respectively, were affected by sarcopenia. Beaudart C et al. [32] evaluated the short-, middle- and long-term health outcomes of sarcopenia in a meta-analysis and found that sarcopenia was associated with higher mortality, functional decline, falls, and hospitalization rates. 

## 4. Discussion

### 4.1. General Knowledge Structure in Sarcopenia Research

Over the past two decades, annual publications of sarcopenia research have grown rapidly, peaking in 2021. As can be seen, sarcopenia is gaining increasing attention among researchers, and the literature in this field is likely to continue to grow.

Among the high-impact countries, the USA occupied the absolute advantage position with the highest number of publications and citations and had a strong collaboration with other countries, such as China, Italy, and the UK. At the same time, Asian countries such as China, Japan, and South Korea also contributed great amounts of publications in this field and their AAY are later, suggesting that they are paying more and more attention to this field in recent years. Interestingly, after adjusting GDP, South Korea reached the first position with more than 190 publications per trillion GDP. Moreover, a significant positive correlation between GDP and the number of publications was found (Appendix A), suggesting that economics is one of the key factors affecting countries’ productivity. However, the percentage of the old population has a low but significant correlation with countries’ productivity (*r* = 0.338, *p* = 0.003). Previous study also observed that countries with high rates of population aging, such as Japan and countries in Europe, are also big producers in research regarding very old populations [49].

The Catholic University of the Sacred Heart in Italy was identified as a leading institution, with the most publications and citations, and it also has a wide collaboration with other institutions. It is also worth noting that four of the top ten most productive institutions are from Asia, including Sichuan University in China, the National Center for Geriatrics and Gerontology in Japan, Yonsei University, and Seoul National University in South Korea. However, their TLS are relatively low, especially the TLS of Sichuan University, which is only 1, which means they were lacking cross-institution collaborations. From Figure 3, the collaborations were mainly restricted to the domestic level, and this situation is also found in other research fields, suggesting that international cooperation needs to be strengthened in the future. Productive authors were mainly from prolific institutions and work closely together. Among the top ten most productive authors, four are from the Catholic University of the Sacred Heart. Landi F contributed most articles and gained the most citations in this field. In addition to participating in reaching the European consensus [1,13], he and his team also analyzed the prevalence of sarcopenia among older adults and elderly nursing home residents and found that it was associated with an increased risk of anorexia, falls, and death [33,34,35,50,51]. Marzetti E, also an influential author at the Catholic University of the Sacred Heart, focused on the pathophysiological mechanism of sarcopenia, arguing that the apoptosis of skeletal muscle cells caused by mitochondrial dysfunction is a major contributor to muscle degeneration in the elderly, which could be a potential therapeutic target [52,53]. From Figure 4, authors from the same country or even the same institution have close cooperation with each other, and the collaboration is mainly concentrated in the western country. A dual-map overlay shows sarcopenia research are mainly published in the disciplines of molecular biology, immunology, and clinical medicine (Figure 5). The core journals identified include *Journal of Cachexia Sarcopenia and Muscle*, *Journal of Nutrition Health and Aging*, and *Nutrients*. Identification of core journals helps researchers to select appropriate journals for reading and publishing their work. Interestingly, Yuan D et al. [20] found that *Osteoporosis international* published the most papers in the sarcopenia field; this inconsistency may be due to the fact that many meeting abstracts were published in this journal, but only articles and reviews were analyzed in our study.

### 4.2. Main Research Domains in Sarcopenia Research

Keywords are carefully selected to represent the topic and core content of a paper, and the co-occurrence relationship and the AAY of keywords are important indicators that reflect the hot topics and development trends of a research field. Further, analysis of co-cited documents can be used to reveal the intellectual structure and thematic clusters [54]. Based on the keywords and co-cited references analysis, the sarcopenia research can be roughly divided into the following aspects: (1) Definition and diagnosis; (2) epidemiology; (3) etiology and pathogenesis; (4) treatment.

#### 4.2.1. Definition and Diagnosis

The definition and diagnosis criteria change as research develops. Currently, a diagnosis of sarcopenia requires a loss of muscle mass combined with a decrease in muscle strength or physical function. Although muscle mass is a major determinant of muscle strength, inconsistency between the two arise from a variety of causes, including adipose or collagen infiltration and decreased neuromuscular function [36]. Moreover, reduction in muscle mass alone is not a good predictor of clinical outcomes [13]. The Asian consensus recommends setting the age cutoff at 60 or 65 [12], emphasizing that this is an age-related muscle disorder, whereas the European consensus does not limit age [1]. At present, many tools have been developed to aid in the diagnosis of sarcopenia. Cluster#3 sarc-f represents a hot spot in diagnostic research. SARC-F is a simple questionnaire that is also easy to implement in the community and consists of strength, assistance walking, rising from a chair, climbing stairs, and falls. It has been proven to be a reliable tool for the detection of sarcopenia patients [55] and is recommended for case finding in clinical practice [1,12]. Moreover, it has also been reported that the combined measurement of calf circumference based on SARC-F can significantly improve diagnostic accuracy [56,57]. Grip strength and gait speed are the most common and convenient methods for assessing muscle strength and physical performance, which is consistent with keyword frequency analysis. Magnetic resonance imaging (MRI) and computed tomography (CT) are considered the gold standard when measuring muscle mass, while dual-energy X-ray absorptiometry (DXA) and bioelectrical impedance analysis (BIA) are more frequently used. The AAY of the keyword “DXA” is relatively early. DXA measures muscle mass by calculating the sum of non-bone and non-fat mass, while it is easily affected by body thickness and hydration status [58]. Additionally, there may be inconsistencies between brands and instruments, which also limits their use [59,60]. The keyword “BIA” has an AAY later than “DXA”. BIA estimates muscle mass based on differences in electrical conductivity of body components. Similar to DXA, several factors affect the accuracy of its results, and the estimation equations/algorithm are device-specific and should be validated against MRI or DXA before use [61]. However, BIA remains a portable, low-cost, and easy-to-use tool that is valuable for epidemiological, clinical, and follow-up studies, and cut-off points have been provided in recent consensus [1,12]. Furthermore, other tools such as ultrasound [62], dilution of D_3_-Creatine [63], and biomarkers [64] have not yet been used in clinical practice. In the future, diagnostic tools and the determination of a cut-off point will remain critical and difficult, and accuracy and feasibility need to be considered.

#### 4.2.2. Epidemiology

The prevalence of sarcopenia varies by diagnostic criteria, ethnicity, and population. In addition to the aforementioned epidemiology of sarcopenia in the general population, sarcopenia in patients with chronic diseases or cancer is also a research hotspot. As shown in Figure 7A, “#1 liver cirrhosis”, “#6 preoperative sarcopenia” and “#8 gastric cancer” were identified as subfields of sarcopenia. Sarcopenia is very common in patients with chronic liver disease, occurs in 30–70% of patients with liver cirrhosis [65], and is thought to be associated with quality of life, hepatic encephalopathy, post-liver transplant outcomes, and even mortality [66,67]. Other chronic diseases, such as chronic obstructive pulmonary disease (COPD) [8,68], chronic kidney disease (CKD) [69,70], and diabetes [71,72] are also independently associated with sarcopenia. Sarcopenia is also of increasing interest in oncology research because of its high prevalence and adverse outcomes. Due to the various definitions used, the prevalence of sarcopenia in patients with different tumors varies from 5–89% [73]. In a recent umbrella review, sarcopenia was found to be associated with poor prognosis in gastric cancer, hepatocellular cancer, urothelial cancer, head and neck cancer, hematological malignancy, pancreatic cancer, breast cancer, colorectal cancer, lung cancer, esophageal cancer, and ovarian cancer, while significantly increasing the risk of postoperative complications and prolonged hospitalization in patients with digestive cancer [74]. In addition, sarcopenia is related to reduced patient response to chemotherapy drugs and increased toxicity, which in turn can lead to further exacerbation of sarcopenia [75].

#### 4.2.3. Etiology and Pathogenesis

The research direction mainly focuses on “satellite cell”, “oxidative stress”, “myocyte apoptosis”, etc. Sarcopenia is characterized by a reduction in the number and size of muscle fibers, and aging, chronic inflammation, endocrine changes, inactivity, nutritional deficiencies, and decreased neuromuscular function are thought to contribute to the development of sarcopenia [76]. Furthermore, these factors are often intertwined and work together. Apoptosis of muscle cells leads to muscle atrophy, but the specific mechanism remains unclear. The dysfunction of mitochondria may play a critical role in promoting apoptosis of muscle cells [53,77]. The proteostasis, biogenesis, dynamics, and autophagy constitute an important part of mitochondrial quality control, and loss of balance will aggravate muscle atrophy, limiting muscle strength and function [78]. Satellite cells are endogenous muscle stem cells and are considered a reservoir for muscle cells. They are activated when muscles are stressed or traumatized, and undergo proliferation and differentiation, participating in the maintenance, growth, and repair of muscle fibers [79]. Age-related reduction in the number and function of satellite cells may be an important reason for the development of sarcopenia [80,81]. Unlike satellite cells, which can only undergo myogenic differentiation, Muscle-derived stem cells (MDSCs) are thought to differentiate into muscular, vascular, nerve, and bone lineage cells [82]. Adipogenesis of MDSCs may cause fat infiltration in skeletal muscle, leading to the loss of muscle strength and promoting the occurrence and development of sarcopenia [83].

#### 4.2.4. Treatment

Since the pathogenesis and molecular mechanism are not fully understood, treatments for sarcopenia are very limited. Exercise and nutritional supplements are widely accepted as effective and safe interventions [12,84]. However, there are various ways of exercise and nutritional supplementation, but there is no unified standard for evaluating efficacy. A meta-analysis compared resistance training (RT), whole-body vibration training (WBVT), and mixed training, and found that all of them increased the time up and go (TUG) times, but only RT and mixed training improved knee extension strength and gait speed [85]. Adequate intake of protein, vitamin D, antioxidant nutrients, and long-chain polyunsaturated fatty acids has a key role in the prevention and treatment of sarcopenia [86]. Supplementation with branched-chain amino acids such as leucine, isoleucine, and valine has been proven effective to reduce muscle loss and improve physical performance [86,87]. Beta-hydroxy-beta-methylbutyrate (HMB) as an active leucine metabolite, stimulates muscle protein synthesis and inhibits protein degradation, and has the potential to prevent and reverse sarcopenia with good safety [88,89]. Ideal pharmacotherapy is still lacking; testosterone and selective androgen receptor modulation (SARM) are the most promising candidates [90], and other possible drugs include growth hormone/insulin growth factor-1 [86], myostatin/activin II receptor inhibitors [87,88], and histone deacetylases (HDACs) [89], but further clinical trials are urgently needed into the efficacy and safety.

### 4.3. Future Trends

Timeline-view analysis and burst detection of cited papers were used to recognize the frontier content and reveal future trends. From Figure 7B, we found that diagnosis tools, clinical characteristics of patients with sarcopenia, and associations between sarcopenia and other diseases were hotspots in recent years. With the newest diagnostic criteria established [1], more studies may continue to focus on the prevalence and clinical outcomes of sarcopenia. However, studies on the pathogenesis and molecular mechanism of sarcopenia seem to have received less attention in the last 5 years, although this might be because our search strategy may have ignored literature on muscle metabolism but without the term “sarcopenia”. At present, the pathogenesis and treatment of sarcopenia remain largely unknown, and the ideal drug therapy is still a long way off. As more and more studies elucidate the harm and severity of sarcopenia, future research needs to pay more attention to the specific mechanisms of muscle deterioration with aging or other disease states and develop corresponding safety and efficacy treatment strategies to prevent sarcopenia and improve the prognosis of patients with sarcopenia.

### 4.4. Limitations

There are still some limitations in our study. First, the definition of sarcopenia was firstly introduced by Irwin Rosenberg in 1989 [91]; however, with the limitation of publications before 1999 that were not collected in the SCIE database, publications between 1989 and 1999 were missing from our study. This may lead us to omit authors and literature who critically contributed to the early stages of sarcopenia development. We further searched references containing “sarcopenia” in their titles in all other databases of WOS and a total of 45 papers were found. Among them, the most cited paper is “Epidemiology of sarcopenia among the elderly in New Mexico” [92] with more than 2500 citations, which estimated the prevalence of sarcopenia in the elderly and analyzed the correlation between low muscle mass and functional impairment, and “Sarcopenia: Origins and Clinical Relevance” [93] which ranks second with more than 1200 citations. Even though we did not conduct a detailed bibliometric analysis of the literature in this period, we believe that the significance of sarcopenia was gradually being recognized and researchers started to explore the prevalence of sarcopenia in the elderly and the criteria of diagnosis.

Second, the bibliometric records retrieved are only from the WOSCC SCI-E database, and we only analyzed the publications with the term “sarcopenia” in the title, which we believed was the most core content in this field. However, this may lead to some missing documents related to the etiology or pathogenesis of sarcopenia but without use of the term “sarcopenia”, especially previous papers discussing the association between muscle loss and chronic wasting diseases. Furthermore, some recently published high-impact publications may be undervalued because they have not accumulated enough citations.

## 5. Conclusions

This current study contributes to advancing knowledge of the research in the sarcopenia field. Bibliometric analysis of the literature showed an increasing effort put in this area and identified the major contributors. Future trends were also revealed through keywords and co-cited references. Pathogenic mechanisms and interventions with efficacy and safety remain a research frontier and critical in the near future.

In summary, our study provides a comprehensive landscape of the evolution process and identified the key characteristics of sarcopenia research in the past 20 years, which may help researchers better understand the knowledge base and future trends in the sarcopenia field.

## Figures and Tables

**Figure 1 ijerph-19-08866-f001:**
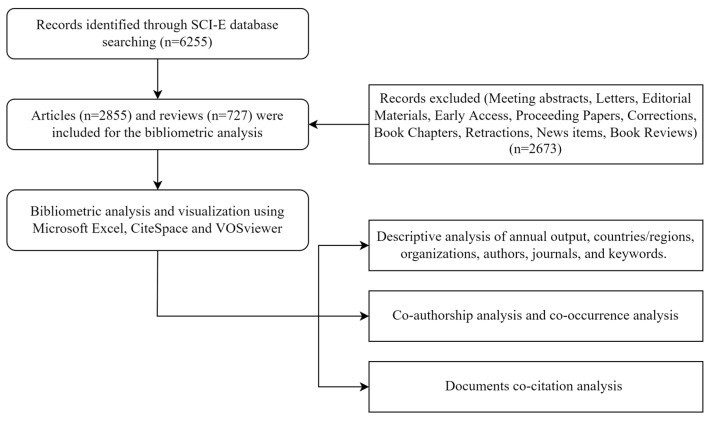
Flow diagram of the literature search, selection, and analysis process.

**Figure 2 ijerph-19-08866-f002:**
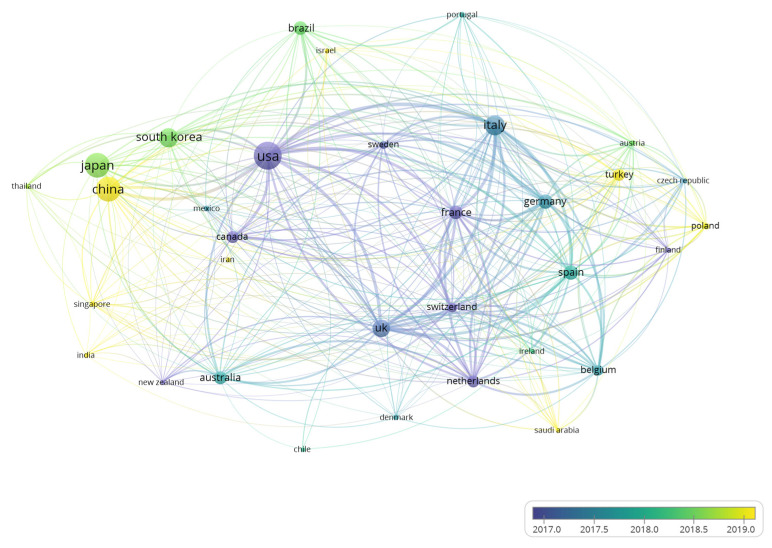
Coauthorship of 34 countries with at least 20 publications in sarcopenia. The size of nodes is determined by the publication numbers. The color indicates the average appearing year (AAY) and the thickness of the lines indicate the strength of the coauthorship.

**Figure 3 ijerph-19-08866-f003:**
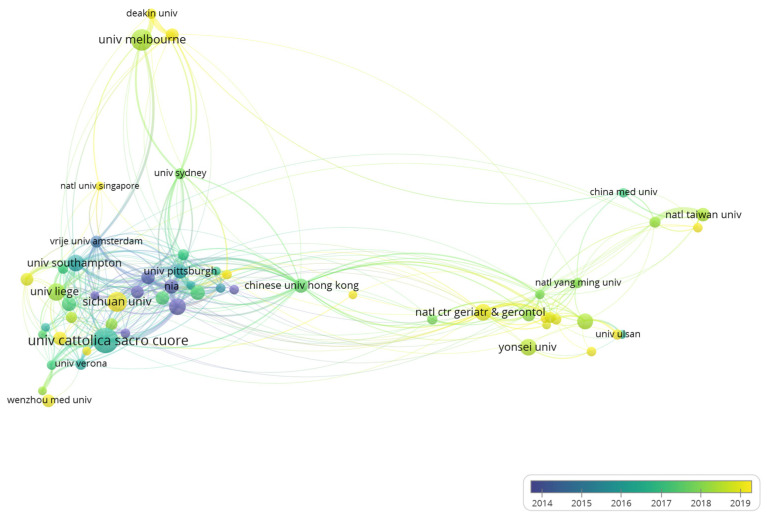
Coauthorship of 58 organizations with at least 20 publications in sarcopenia.

**Figure 4 ijerph-19-08866-f004:**
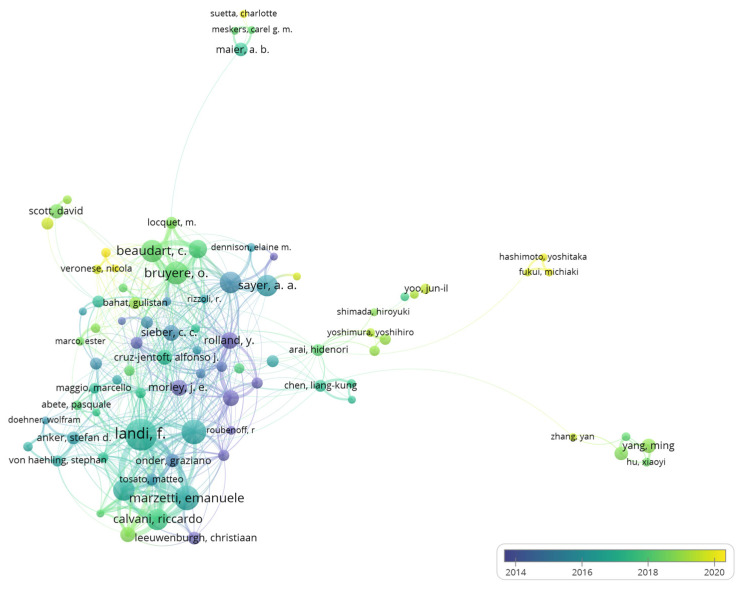
Coauthorship of 85 authors with at least 10 publications in sarcopenia.

**Figure 5 ijerph-19-08866-f005:**
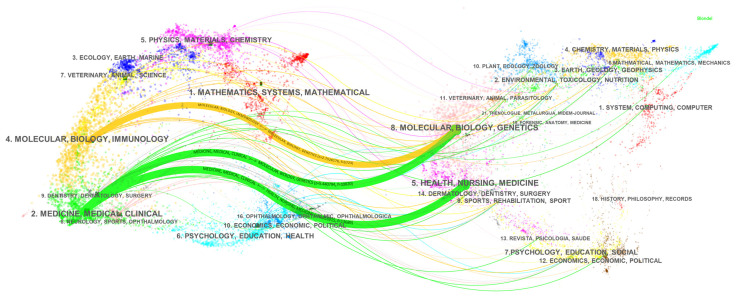
The dual-map overlay of citing journals on the left side and cited journals on the right.

**Figure 6 ijerph-19-08866-f006:**
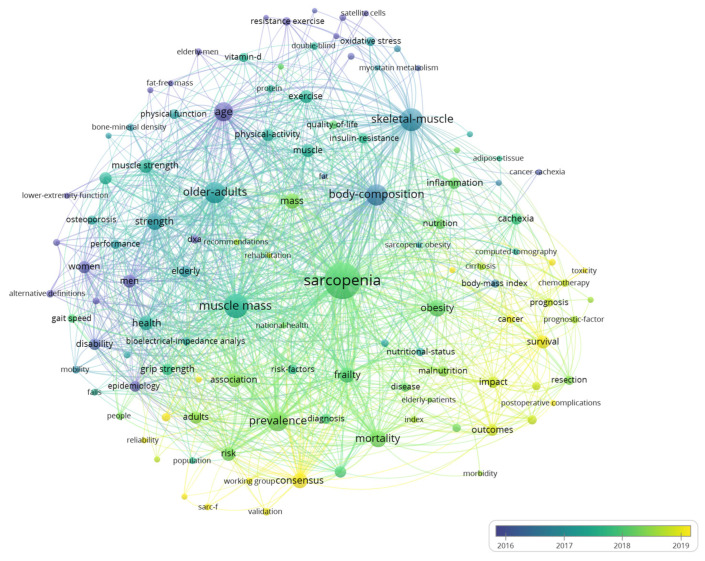
Co-occurrence analysis of 116 keywords with at least 50 times of occurrence.

**Figure 7 ijerph-19-08866-f007:**
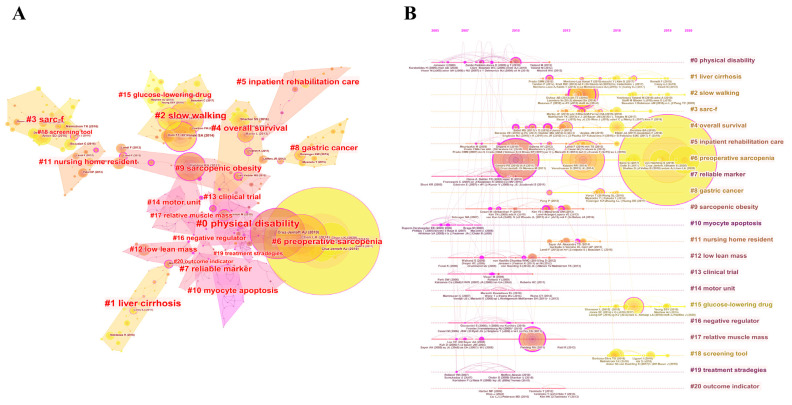
(**A**) The cluster view of the document co-citation network, generated by the top 50 documents per slice between 2010 to 2021. (**B**) The timeline view of the document co-citation network.

**Figure 8 ijerph-19-08866-f008:**
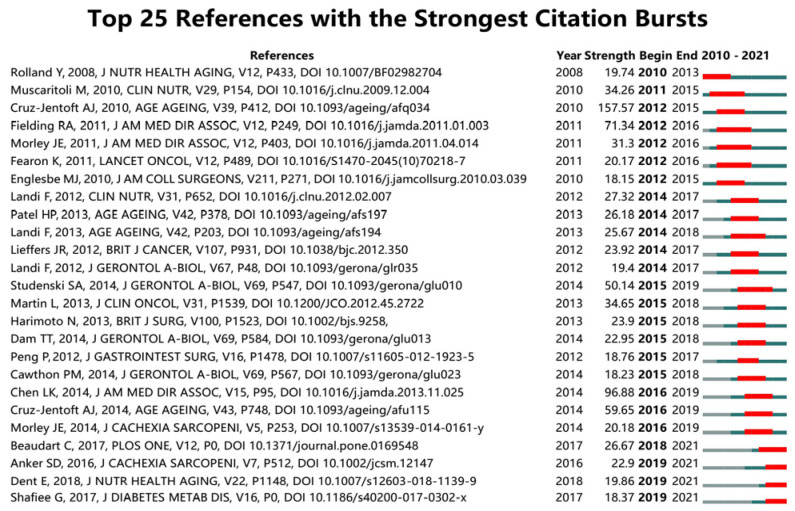
The 25 references with strongest citation bursts between 2010 to 2021; the red bars represent the duration of bursts [9,10,11,13,14,29,30,31,32,33,34,35,36,37,38,39,40,41,42,43,44,45,46,47,48].

**Table 1 ijerph-19-08866-t001:** The top 10 most active journals in the sarcopenia research field.

Journals	Documents	Citations	IF ^1^
Journal of Cachexia Sarcopenia and Muscle	117	4208	12.91
Journal of Nutrition Health and Aging	108	3868	4.075
Nutrients	100	1337	5.719
Journal of the American Medical Directors Association	90	9931	4.669
Journals of Gerontology Series A: Biological Sciences and Medical Sciences	79	6968	6.053
Experimental Gerontology	73	2857	4.032
Clinical Nutrition	71	3876	7.325
Aging Clinical and Experimental Research	68	2017	3.638
Plos One	66	2645	3.24
Geriatrics and Gerontology International	65	2352	2.73

^1^ IF refers to the 2020 impact factor obtained from Journal Citation Reports (Clarivate Analytics, Philadelphia, PA, USA).

## Data Availability

The datasets used and/or analyzed during the current study are available from the corresponding author on reasonable request.

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
