# Peer review of "Bibliometric Analysis of the Knowledge Base and Future Trends on Sarcopenia from 1999–2021"

_ijerph, 2022, doi:10.3390/ijerph19148866_

Round 1
Reviewer 1 Report
This is a very interesting review with a very unique perspective. The co-occurrence analysis in Figure 4 is of particular value. However, I think there are some points that need to be fixed.
Sarcopenia can be divided into primary sarcopenia (age-related weakness) and secondary sarcopenia (disease-induced sarcopenia), which will be treated together in this study. In recent years, there has been less division, which can be a problem when evaluating previous papers. P9, Line274-6 Sarcopenia has been diagnosed by Asian and European consensus on diagnostic criteria, so I thought it would be good to mention this. The resolution of Figur3 and Figur4 is low, and the text is difficult to read.
Reviewer 2 Report
In the present article the Authors report a bibliometric analysis of the published research on sarcopenia from 1999-2021.
Overall, the article is informative because it provides a picture of the literature on sarcopenia and how it has been expanding in the last years, and it provides a detailed overview of the more influent Researchers and Institutions in the field. The article provides several interesting and detailed data.
However, it needs some refinements that can further improve the reading.
Line 46-47: And a landmark…. The sentence is not clear.
Line 112: 79 countries/regions…..The sentence is not clear.
Line 137: There are 98 authors… The sentence is incorrect.
Line 167-168: Except for sarcopenia….The sentence is incorrect and should be rephrased to provide a clear concept.
Line 182-183: The European consensus…The sentence is incorrect.
Line 198-201: The sentence is incorrect.
Line 306: the homeostasis of proteostasis is incorrect, maybe is sufficient to speak about proteostasis.
Major points:
My major concern is lack of information in the period 1989-1999. In fact, the first definition of Sarcopenia by Rosenberg dates back to 1989 (Rosenberg. Summary comments. Am J Clin Nutr. 1989; (50):1231–3.). I think that Authors should mention and comment the literature trend in the period 1989-1999. Even though it is not in the scope of the present analysis it could provide interesting information and further strengthen the reported observations. In this context, the sentence in lines 175-177 (probably because it revealed….) is incorrect and should be rephrased in the view of this comment.
In the title Authors mention the “future trends”. I think Authors should better develop how They see the future development of the literature on sarcopenia. For example, in lines 190-192 Authors report a shift on research trends and hot spots. I think that Researchers shifted their attention from the initial observation of a macroscopic evidence such as physical inability, to the search of molecular mechanisms and to the evidence that sarcopenia is not only related to aged people but also in multiple events of life. I suggest the introduction of a dedicated paragraph in the Discussion to explain the possible future trends, and/or how the knowledge has influenced the fight against sarcopenia.
The conclusions paragraph seems to me a repetition of the Discussion. Maybe it is better to mention only the summarizing sentences.
Figures 3, 4, and 5 are too small and the quality should be improved; in Figure 3D and 5B is not possible to read the letters even at higher magnification on the computer screen.
Reviewer 3 Report
First of all, I would like to thank the journal IJERPH for the opportunity to review this article. Bibliometric analysis helps researchers and clinicians to identify emerging and innovative topics, in this case in the area of sarcopenia. Moreover, the authors have shown a great amount of work in this manuscript. However, for publication I suggest a major review in order to improve its quality and also an extensive English language spell check.
Specific comments:
INTRODUCTION:
- In the lines 53 to 58 the authors name and state other bibliometric studies related to sarcopenia, please clarify what the presented study adds to the literature and why another bibliometric study is “urgently needed” (line 61-62).
- Please add a clearer objective to the study since there are a couple of sentences but they don’t define clearly (under my point of view) the objective of the study.
METHODS:
- Under my opinion this section could benefit of subtitles such as: database selection, search strategy, Standardization of the bibliographic information and analysis of the scientific production (bibliometric indicators).
- Please explain more thoroughly the standardization process of the different names of a particular author name (eg. sometimes an author appears as 2 or more signatures) and institutions (eg. the different variants of university or hospital names).
RESULTS: very interesting, but to highlight the results, under my opinion the four figures of Figure 3 should be presented as separate figures (also to improve quality of the image), as well as Figure 4A and the figures in Figure 5. I understand you may have a limit, but figure 2 and figure 7 are not necessary since that information is in the text as well as Figure 4 B which could be eliminated.
DISCUSSION: this is the section that needs more changes. The discussion that is presented is mostly related to keywords but the results show information related to authorship, instititions or countries' productivity, and about citation. I think authors should add more discussion about these bibliometric indicators. Also I would add some more discussion regarding previous bibliometric studies related to sarcopenia or aging and if the most productive authors, countries and institutions also seem to be the most productive in other related studies, as well as more discussion about citation results and the timeline. For example, regarding countries does productivity it depend on aged population or gross domestic product (GDP)?
CONCLUSION: unless the discussion is changed the conclusions don’t make sense since the information about countries and authors is not discussed in the present form.
Round 2
Reviewer 2 Report
Authors have answered all the raised points.
Reviewer 3 Report
Thank you for considering my comments and the ones of the other reviewers. I think overall it is a very interesting manuscript.